# From which world is your graph?

**Cheng Li**
College of William & Mary

**Felix M. F. Wong**
Independent Researcher*

**Zhenming Liu**
College of William & Mary

**Varun Kanade**
University of Oxford

## Abstract

Discovering statistical structure from links is a fundamental problem in the analysis of social networks. Choosing a misspecified model, or equivalently, an incorrect inference algorithm will result in an invalid analysis or even falsely uncover patterns that are in fact artifacts of the model. This work focuses on unifying two of the most widely used link-formation models: the stochastic blockmodel (SBM) and the small world (or latent space) model (SWM). Integrating techniques from kernel learning, spectral graph theory, and nonlinear dimensionality reduction, we develop the first statistically sound polynomial-time algorithm to discover latent patterns in sparse graphs for both models. When the network comes from an SBM, the algorithm outputs a block structure. When it is from an SWM, the algorithm outputs estimates of each node's latent position.

## 1 Introduction

Discovering statistical structures from links is a fundamental problem in the analysis of social networks. Connections between entities are typically formed based on underlying feature-based similarities; however these features themselves are partially or entirely hidden. A question of great interest is to what extent can these latent features be inferred from the observable links in the network. This work focuses on the so-called assortative setting, the principle that *similar individuals are more likely to interact with each other*. Most stochastic models of social networks rely on this assumption, including the two most famous ones – the stochastic blockmodel [1] and the small-world model [2, 3], described below.

**Stochastic Blockmodel (SBM)**. In a stochastic blockmodel [4, 5, 6, 7, 8, 9, 10, 11, 12, 13], nodes are grouped into disjoint "communities" and links are added randomly between nodes, with a higher probability if nodes are in the same community. In its simplest incarnation, an edge is added between nodes within the same community with probability $p$, and between nodes in different communities with probability $q$, for $p > q$. Despite arguably naïve modelling choices, such as the independence of edges, algorithms designed with SBM work well in practice [14, 15].

**Small-World Model (SWM)**. In a small-world model, each node is associated with a latent variable $x_i$, *e.g.,* the geographic location of an individual. The probability that there is a link between two nodes is proportional to an inverse polynomial of some notion of distance, $\mathrm{dist}(x_i, x_j)$, between them. The presence of a small number of "long-range" connections is essential to some of the most intriguing properties of these networks, such as small diameter and fast decentralized routing algorithms [3]. In general, the latent position may reflect geographic location as well as more abstract concepts, *e.g.,* position on a political ideology spectrum.

**The Inference Problem**. Without observing the latent positions, or knowing which model generates the underlying graph, the adjacency matrix of a social graph typically looks like the one shown in

Fig. 5(a) (App. A.1). However, if the model generating the graph is known, it is then possible to run a suitable "clustering algorithm" [14, 16] that reveals the hidden structure. When the vertices are ordered suitably, the SBM's adjacency matrix looks like the one shown in Fig. 5(b) (App. A.1) and that of the SWM looks like the one shown in Fig. 5(c) (App. A.1). Existing algorithms typically depend on knowing the "true" model and are tailored to graphs generated according to one of these models, *e.g.,* [14, 16, 17, 18].

**Our Contributions**. We consider a latent space model that is general enough to include both these models as special cases. In our model, an edge is added between two nodes with a probability that is a decreasing function of the distance between their latent positions. This model is a fairly natural one, and it is quite likely that a variant has already been studied; however, to the best of our knowledge there is no known statistically sound and computationally efficient algorithm for latent-position inference on a model as general as the one we consider.

*1. A unified model.* We propose a model that is a natural generalization of both the stochastic blockmodel and the small-world model that captures some of the key properties of real-world social networks, such as small out-degrees for ordinary users and large in-degrees for celebrities. We focus on a simplified model where we have a modest degree graph only on "celebrities"; the full paper material contains an analysis of the more realistic model using somewhat technical machinery [19].

*2. A provable algorithm.* We present statistically sound and polynomial-time algorithms for inferring latent positions in our model(s). Our algorithm approximately infers the latent positions of almost all "celebrities" $(1 - o(1)$-fraction), and approximately infers a constant fraction of the latent positions of ordinary users. We show that it is statistically impossible to err on at most $o(1)$ fraction of ordinary users by using standard lower bound arguments.

*3. Proof-of-concept experiments.* We report several experiments on synthetic and real-world data collected on Twitter from Oct 1 and Nov 30, 2016. Our experiments demonstrate that our model and inference algorithms perform well on real-world data and reveal interesting structures in networks.

**Additional Related Work**. We briefly review the relevant published literature. *1. Graphon & Latent-space techniques.* Studies using graphons and latent-space models have focused on the statistical properties of the estimators [20, 21, 22, 23, 24, 25, 26, 27, 28], with limited attention paid to computational efficiency. The "USVT" technique developed recently [29] estimates the kernel well when the graph is dense. Xu et al. [30] consider a polynomial time algorithm for a sparse model similar to ours, but focus on edge classification rather than latent position estimation. *2. Correspondence analysis in political science.* Estimating the ideology scores of politicians is an important research topic in political science [31, 32, 33, 34, 35, 36, 17, 18]. High accuracy heuristics developed to analyze dense graphs include [17, 18].

**Organization**. Section 2 describes background, our model and results. Section 3 describes our algorithm and an gives an overview of its analysis. Section 4 contains the experiments.

## 2 Preliminaries and Summary of Results

**Basic Notation**. We use $c_0, c_1$, etc. to denote constants which may be different in each case. We use whp to denote with high probability, by which we mean with probability larger $1 - \frac{1}{n^c}$ for any $c$. All notation is summarized in Appendix B for quick reference.

**Stochastic Blockmodel**. Let $n$ be the number of nodes in the graph with each node assigned a label from the set $\{1, \ldots, k\}$ uniformly at random. An edge is added between two nodes with the same label with probability $p$ and between the nodes with different labels with probability $q$, with $p > q$ (assortative case). In this work, we focus on the $k = 2$ case, where $p, q = \Omega\left((\log n)^c/n\right)$ and the community sizes are exactly the same. (Many studies of the regimes where recovery is possible have been published [37, 9, 5, 8].)

Let $A$ be the adjacency matrix of the realized graph and let $M = \mathbb{E}[A] = \begin{pmatrix} P & Q \\ Q & P \end{pmatrix}$, where $P$ and $Q \in R^{\frac{n}{2} \times \frac{n}{2}}$ with every entry equal to $p$ and $q$, respectively. We next explain the inference algorithm, which uses two key observations. *1. Spectral Properties of $M$.* $M$ has rank 2 and the non-trivial eigenvectors are $(1, \ldots, 1)^{\mathsf{T}}$ and $(1, \ldots, 1, -1, \ldots, -1)$ corresponding to eigenvalues $n(p+q)/2$ and $n(p-q)/2$, respectively. If one has access to $M$, the hidden structure in the graph is revealed merely by reading off the second eigenvector. *2. Low Discrepancy between $A$ and*

$M$. Provided the average degree $n(p+q)/2$ and the gap $p-q$ are large enough, the spectrum and eigenspaces of the matrices $A$ and $M$ can be shown to be close using matrix concentration inequalities and the Davis-Kahan theorem [38, 39]. Thus, it is sufficient to look at the projection of the columns of $A$ onto the top two eigenvectors of $A$ to identify the hidden latent structure.

**Small-World Model (SWM)**. In a 1-dim. SWM, each node $v_i$ is associated with an independent latent variable $x_i \in [0,1]$ that is drawn from the uniform distribution on $[0,1]$. The probability of a link between two nodes is $\Pr[\{v_i, v_j\} \in E] \propto \frac{1}{|x_i - x_j|^\Delta + c_0}$, where $\Delta > 1$ is a hyper-parameter.

The inference algorithm for small-world models uses different ideas. Each edge in the graph is considered as either "short-range" or "long-range." Short-range edges are those between nodes that are nearby in latent space, while long-range edges have end-points that are far away in latent space. After removing the long-range edges, the shortest path distance between two nodes scales proportionally to the corresponding latent space distance (see Fig. 6 in App. A.2). After obtaining estimates for pairwise distances, standard buidling blocks are used to find the latent positions $x_i$ [40]. The key observation used to remove the long-range edges is: an edge $\{v_i, v_j\}$ is a short-range edge if and only if $v_i$ and $v_j$ will share many neighbors.

**A Unified Model**. Both SBM and SWM are special cases of our unified latent space model. We begin by describing the full-fledged bipartite (heterogeneous) model that is a better approximation of real-world networks, but requires sophisticated algorithmic techniques (see [19] for a detailed analysis). Next, we present a simplified (homogeneous) model to explain the key ideas.

*Bipartite Model*. We use latent-space model to characterize the stochastic interactions between users. Each individual is associated with a latent variable in $[0,1]$. The bipartite graph model consists of two types of users: the left side of the graph $\mathbf{Y} = \{y_1, \ldots, y_m\}$ are the *followers* (ordinary users) and the right side $\mathbf{X} = \{x_1, \ldots, x_n\}$ are the *influencers* (celebrities). Both $y_i$ and $x_i$ are i.i.d. random variables from a distribution $\mathcal{D}$. This assumption follows the convention of existing heterogeneous models [41, 42]. The probability that two individuals $y_i$ and $x_j$ interact is $\kappa(y_i, x_j)/n$, where $\kappa : [0,1] \times [0,1] \to (0,1]$ is a kernel function. Throughout this paper we assume that $\kappa$ is a small-world kernel, *i.e.,* $\kappa(x,y) = c_0/(\|x - y\|^\Delta + c_1)$ for some $\Delta > 1$ and suitable constants $c_0, c_1$, and that $m = \Theta(n \cdot \text{polylog}(n))$. Let $B \in R^{m \times n}$ be a binary matrix that $B_{i,j} = 1$ if and only if there is an edge between $y_i$ and $x_j$. Our goal is to estimate $\{x_i\}_{i \in [n]}$ based on $B$ for suitably large $n$.

*Simplified Model*. The graph only has the node set is $\mathbf{X} = \{x_1, ..., x_n\}$ of celebrity users. Each $x_i$ is again an i.i.d. random variable from $\mathcal{D}$. The probability that two users $v_i$ and $v_j$ interact is $\kappa(x_i, x_j)/C(n)$. The denominator is a normalization term that controls the edge density of the graph. We assume $C(n) = n/\text{polylog}(n)$, *i.e.,* the average degree is $\text{polylog}(n)$. Unlike the SWM where the $x_i$ are drawn uniformly from $[0,1]$, in the unified model $\mathcal{D}$ can be flexible. When $\mathcal{D}$ is the uniform distribution, the model is the standard SWM. When $\mathcal{D}$ has discrete support (*e.g.,* $x_i = 0$ with prob. 1/2 and $x_i = 1$ otherwise), then the unified model reduces to the SBM. Our distribution-agnostic algorithm can automatically select the most suitable model from SBM and SWM, and infer the latent positions of (almost) all the nodes.

**Bipartite vs. Simplified Model**. The simplified model suffers from the following problem: If the average degree is $O(1)$, then we err on estimating every individual's latent position with a constant probability (*e.g.,* whp the graph is disconnected), but in practice we usually want a high prediction accuracy on the subset of nodes corresponding to high-profile users. Assuming that the average degree is $\omega(1)$ mismatches empirical social network data. Therefore, we use a bipartite model that introduces heterogeneity among nodes: By splitting the nodes into two classes, we achieve high estimation accuracy on the influencers and the degree distribution more closely matches real-world data. For example, in most online social networks, nodes have $O(1)$ average degree, and a small fraction of users (influencers) account for the production of almost all "trendy" content while most users (followers) simply consume the content.

**Additional Remarks on the Bipartite Model**. *1. Algorithmic contribution*. Our algorithm computes $B^\mathsf{T} B$ and then regularizes the product by shrinking the diagonal entries before carrying out spectral analysis. Previous studies of the bipartite graph in similar settings [43, 44, 45] attempt to construct a regularized product using different heuristics. Our work presents the first theoretically sound regularization technique for spectral algorithms. In addition, some studies have suggested running SVD on $B$ directly (*e.g.,* [28]). We show that the (right) singular vectors of $B$ *do not* converge

to the eigenvectors of $K$ (the matrix with entries $\kappa(x_i, x_j)$). Thus, it is necessary to take the product and use regularization. *2. Comparison to degree-corrected models (DCM)*. In DCM, each node $v_i$ is associated with a degree parameter $D(v_i)$. Then we have $\Pr[\{v_i, v_j\} \in \mathbb{E}] \propto D(v_i)\kappa(x_i, x_j)D(v_j)$. The DCM model implies the subgraph induced by the highest degree nodes is dense, which is inconsistent with real-world networks. There is a need for better tools to analyze the asymptotic behavior of such models and we leave this for future work (see, *e.g.,* [41, 42]).

**Theoretical Results**. Let $F$ be the cdf of $\mathcal{D}$. We say $F$ and $\kappa$ are **well-conditioned** if:
*(1) $F$ has finitely many points of discontinuity*, *i.e.,* the closure of the support of $F$ can be expressed as the union of non-overlapping closed intervals $I_1, I_2, ..., I_k$ for a finite number $k$.
*(2) $F$ is near-uniform*, *i.e.,* for any interval $I$ that has non-empty overlap with $F$'s support, $\int_I dF(x) \geq c_0|I|$, for some constant $c_0$.
*(3) Decay Condition: The eigenvalues of the integral operator based on $\kappa$ and $F$ decay sufficiently fast.* We define the $\mathcal{K}f(x) = \int \kappa(x, x')f(x')dF(x')$ and let $(\lambda_i)_{i \geq 1}$ denote the eigenvalues of $\mathcal{K}$. Then, it holds that $\lambda_i = O(i^{-2.5})$.

If we use the small-word kernel $\kappa(x, y) = c_0/(|x - y|^\Delta + c_1)$ and choose $F$ that gives rise to SBM or SWM, in each case the pair $F$ and $\kappa$ are well-conditioned, as described below. As the decay condition is slightly more invoved, we comment upon it. The condition is a mild one. When $F$ is uniformly distributed on $[0, 1]$, it is equivalent to requiring $\mathcal{K}$ to be twice differentiable, which is true for the small world kernel. When $F$ has a finite discrete support, there are only finitely many non-zero eigenvalues, *i.e.,* this condition also holds. The decay condition holds in more general settings, *e.g.,* when $F$ is piecewise linear [46] (see [19]). Without the decay condition, we would require much stronger assumptions: Either the graph is very dense or $\Delta \gg 2$. Neither of these assumptions is realistic, so effectively our algorithm fails to work. In practice, whether the decay condition is satisfied can be checked by making a log-log plot and it has been observed that for several real-world networks, the eigenvalues follow a power-law distribution [47].

Next, we define the notion of latent position recovery for our algorithms.

**Definition 2.1** (($\alpha, \beta, \gamma$)-Aproximation Algorithm). *Let $I_i$, $F$, and $\mathcal{K}$ be defined as above, and let $R_i = \{x_j : x_j \in I_i\}$. An algorithm is called an ($\alpha, \beta, \gamma$)-approximation algorithm if*
*1. It outputs a collection of disjoint points $C_1, C_2, \ldots, C_k$ such that $C_i \subseteq R_i$, which correspond to subsets of reconstructed latent variables.*
*2. For each $C_i$, it produces a distance matrix $D^{(i)}$. Let $G_i \subseteq C_i$ be such that for any $i_j, i_k \in G_i$*

$$D^{(i)}_{i_j, i_k} \leq |x_{i_j} - x_{i_k}| \leq (1 + \beta)D^{(i)}_{i_j, i_k} + \gamma. \tag{1}$$

*3. $|\bigcup_i G_i| \geq (1 - \alpha)n$.*
*In bipartite graphs, Eq.(1) is required only for influencers.*

We do not attempt to optimize constants in this paper. We set $\alpha = o(1)$, $\beta$ a small constant, and $\gamma = o(1)$. Definition 2.1 allows two types of errors: $C_i$s are not required to form a partition *i.e.,* some nodes can be left out, and a small fraction of estimation errors is allowed in each $C_i$, *e.g.,* if $x_j = 0.9$ but $\widehat{x}_j = 0.2$, then the $j$-th "row" in $D^{(i)}$ is incorrect. To interpret the definition, consider the blockmodel with 2 communities. Condition 1 means that our algorithm will output two disjoint groups of points. Each group corresponds to one block. Condition 2 means that there are pairwise distance estimates within each group. Since the true distances for nodes within the same block are zero, our estimates must also be zero to satisfy Eq.1. Condition 3 says that the proportion of misclassified nodes is $\alpha = o(1)$. We can also interpret the definition when we consider a small-world graph, in which case $k = 1$. The algorithm outputs pairwise distances for a subset $C_1$. We know that there is a sufficiently large $G_1 \subseteq C_1$ such that the pairwise distances are all correct in $C_1$.

Our algorithm does not attempt to estimate the distance between $C_i$ and $C_j$ for $i \neq j$. When the support contains multiple disjoint intervals, *e.g.,* in the SBM case, it first pulls apart the nodes in different communities. Estimating the distance between intervals, given the output of our algorithm is straightforward. Our main result is the following.

**Theorem 2.2.** *Using the notation above, assume $F$ and $\kappa$ are well-conditioned, and $C(n)$ and $m/n$ are $\Omega(\log^c n)$ for some suitably large $c$. The algorithm for the simplified model shown in Figure 1 and that for the bipartite model (appears in [19]) give us an $(1/\log^2 n, \epsilon, O(1/\log n))$-approximation algorithm w.h.p. for any constant $\epsilon$. Furthermore, the distance estimates $D^{(i)}$ for each $C_i$ are constructed using the shortest path distance of an unweighted graph.*

LATENT-INFERENCE($A$)

1  // **Step 1. Estimate** $\Phi$ .
2  $\widehat{\Phi} = $ SM-EST($A$).
3  // **Step 2. Execute isomap algo.**
4  $D = $ ISOMAP-ALGO($\widehat{\Phi}$)
5  // **Step 3. Find latent variables.**
6  Run a line embedding algorithm [48, 49].

ISOMAP-ALGO($\widehat{\Phi}, \ell$)

1  Execute $S \leftarrow$ DENOISE($\widehat{\Phi}$) (See Section 3.2)
2  // $S$ is a subset of $[n]$.
3  Build $G = \{S, E\}$ s.t. $\{i, j\} \in E$ iff
4     $|(\tilde{\Phi}_d)_i - (\tilde{\Phi}_d)_j| \leq \ell / \log n$ ($\ell$ a constant).
5  Compute $D$ such $D(i, j)$ is the shortest
6     path distance between $i$ and $j$ when $i, j \in S$.
7  **return** D

SM-EST($A, t$)

1  $[\tilde{U}_A, \tilde{S}_A, \tilde{V}_A] = $ svd($A$).
2  Let also $\lambda_i$ be $i$-th singular value of $A$.
3  // let $t$ be a suitable parameter.
4  $d = $ DECIDETHRESHOLD($t, \rho(n)$).
5  $S_A$: diagonal matrix comprised of $\{\lambda_i\}_{i \leq d}$
6  $U_A, V_A$: the singular vectors
7     corresponding to $S_A$.
8  Let $\widehat{\Phi} = \sqrt{C(n)} U_A S_A^{1/2}$.
9  **return** $\widehat{\Phi}$

DECIDETHRESHOLD($t, \rho(n)$)

1  // This procedure decides $d$ the number
2     of Eigenvectors to keep.
3  // $t$ is a tunable parameter. See Proposition 3.1.
4  $d = \arg\max_d \{\lambda_d(\frac{A}{\rho(n)}) - \lambda_{d+1}(\frac{A}{\rho(n)}) \geq \theta\}$.
5  where $\theta = 10(t/\rho(n))^{24/59}$

Figure 1: Subroutines of our Latent Inference Algorithm.

**Pairwise Estimation to Line-embedding and High-dimensional Generalization.** Our algorithm builds estimates on pairwise latent distance and uses well-studied metric-embedding methods [48, 49] as blackboxes to infer latent positions. Our inference algorithm can be generalized to $d$-dimensional space with $d$ being a constant. But the metric-embedding on $\ell_p^d$ becomes increasingly difficult, *e.g.,* when $d = 2$, the approximation ratio for embedding a graph is $\Omega(\sqrt{n})$ [50].

# 3   Our algorithms

As previously noted, SBM and SWM are special cases of our unified model and both require different algorithmic techniques. Given that it is not surprising that our algorithm blends ingredients from both sets of techniques. Before proceeding, we review basics of kernel learning.

**Notation**. Let $A$ be the adjacency matrix of the observed graph (simplified model) and let $\rho(n) \triangleq n/C(n)$. Let $K$ be the matrix with entries $\kappa(x_i, x_j)$. Let $\tilde{U}_K \tilde{S}_K \tilde{V}_K^{\mathsf{T}}$ $(\tilde{U}_A \tilde{S}_A \tilde{V}_A^{\mathsf{T}})$ be the SVD of $K$ ($A$). Let $d$ be a parameter to be chosen later. Let $S_K$ ($S_A$) be a $d \times d$ diagonal matrix comprising the $d$-largest eigenvalues of $K$ ($A$). Let $U_K$ ($U_A$) and $V_K$ ($V_A$) be the corresponding singular vectors of $K$ ($A$). Finally, let $\bar{K} = U_K S_K V_K^{\mathsf{T}}$ $(\bar{A} = U_A S_A V_A^{\mathsf{T}})$ be the low-rank approximation of $K$ ($A$). Note that when a matrix is positive definite and symmetric SVD coincides with eigen-decomposition; as a consequence $U_K = V_K$ and $U_A = V_A$.

**Kernel Learning.** Define an integral operator $\mathcal{K}$ as $\mathcal{K}f(x) = \int \kappa(x, x')f(x')dF(x')$. Let $\psi_1, \psi_2, \ldots$ be the eigenfunctions of $\mathcal{K}$ and $\lambda_1, \lambda_2, \ldots$ be the corresponding eigenvalues such that $\lambda_1 \geq \lambda_2 \geq \cdots$ and $\lambda_i \geq 0$ for each $i$. Also let $N_{\mathcal{H}}$ be the number of eigenfunctions/eigenvalues of $\mathcal{K}$, which is either finite or countably infinite. We recall some important properties of $\mathcal{K}$ [51, 25]. For $x \in [0, 1]$, define the feature map $\Phi(x) = (\sqrt{\lambda_j}\psi_j(x) : j = 1, 2, \ldots)$, so that $\langle \Phi(x), \Phi(x') \rangle = \kappa(x, x')$. We also consider a truncated feature $\Phi_d(x) = (\sqrt{\lambda_j}\psi_j(x) : j = 1, 2, \ldots, d)$. Intuitively, if $\lambda_j$ is too small for sufficiently large $j$, then the first $d$ coordinates (*i.e.,* $\Phi_d$) already approximate the feature map well. Finally, let $\Phi_d(\mathbf{X}) \in \mathbb{R}^{n \times d}$ such that its $(i, j)$-th entry is $\sqrt{\lambda_j}\psi_j(x_i)$. Let's further write $(\Phi_d(\mathbf{X}))_{:,i}$ be the $i$-th column of $\Phi_d(\mathbf{X})$. Let $\Phi(\mathbf{X}) = \lim_{d \to \infty} \Phi_d(\mathbf{X})$. When the context is clear, shorten $\Phi_d(\mathbf{X})$ and $\Phi(\mathbf{X})$ to $\Phi_d$ and $\Phi$, respectively.

There are two main steps in our algorithm which we explain in the following two subsections.

## 3.1   Estimation of $\Phi$ through $K$ and $A$

The mapping $\Phi : [0, 1] \to \mathbb{R}^{N_{\mathcal{H}}}$ is bijective so a (reasonably) accurate estimate of $\Phi(x_i)$ can be used to recover $x_i$. Our main result is the design of a data-driven procedure to choose a suitable number of eigenvectors and eigenvalues of $A$ to approximate $\Phi$ (see SM-EST($A$) in Fig. 1).

**Proposition 3.1.** *Let $t$ be a tunable parameter such that $t = o(\rho(n))$ and $t^2/\rho(n) = \omega(\log n)$. Let $d$ be chosen by* DECIDETHRESHOLD$(\cdot)$*. Let $\widehat{\Phi} \in \mathbb{R}^{N_{\mathcal{H}}}$ be such that its first $d$-coordinates are equal to $\sqrt{C(n)}U_A S_A^{1/2}$, and its remaining entries are 0. If $\rho(n) = \omega(\log n)$ and $\mathcal{K}$ (F and $\kappa$) is well-conditioned, then with high probability:*

$$\|\widehat{\Phi} - \Phi\|_F = O\left(\sqrt{n}\left(t/(\rho(n))\right)^{\frac{2}{29}}\right) \tag{2}$$

Specifically, by letting $t = \rho^{2/3}(n)$, we have $\|\widehat{\Phi} - \Phi\|_F = O\left(\sqrt{n}\rho^{-2/87}(n)\right)$. We remark that our result is stronger than an analogous result for sparse graphs in [25] as our estimate is close to $\Phi$ rather than the truncated $\Phi_d$.

**Remark on the Eigengap.** In our analysis, there are three groups of eigenvalues: the eigenvalues of $\mathcal{K}$, those of $K$, and those of $A$. They are in different scales: $\lambda_i(\mathcal{K}) \leq 1$ (resulting from the fact that $\kappa(x,y) \leq 1$ for all $x$ and $y$), and $\lambda_i(A/\rho(n)) \approx \lambda_i(K/n) \approx \lambda_i(\mathcal{K})$ if $n$ and $\rho(n)$ are sufficiently large. Thus, $\lambda_d(\mathcal{K})$ are *independent of $n$* for a *fixed $d$* and should be treated as $\Theta(1)$. Also $\delta_d \triangleq \lambda_d(\mathcal{K}) - \lambda_{d+1}(\mathcal{K}) \to 0$ as $d \to \infty$. Since *the procedure of choosing $d$ depends on $C(n)$* (and thus also on $n$), $\delta_d$ depends on $n$ and can be bounded by a function in $n$. This is the reason why Proposition 3.1 does not explicitly depend on the eigengap. We also note that we cannot directly find $\delta_d$ based on the input matrix $A$. But standard interlacing results can give $\delta_d = \Theta(\lambda_d(A/\rho(n)) - \lambda_{d+1}(A/\rho(n)))$ (cf. [19]).

**Intuition of the algorithm.** Using Mercer's theorem, we have $\langle \Phi(x_i), \Phi(x_j) \rangle = \lim_{d \to \infty} \langle \Phi_d(x_i), \Phi_d(x_j) \rangle = \kappa(x_i, x_j)$. Thus, $\lim_{d \to \infty} \Phi_d \Phi_d^{\mathsf{T}} = K$. On the other hand, we have $(\tilde{U}_K \tilde{S}_K^{1/2})(\tilde{U}_K \tilde{S}_K^{1/2})^{\mathsf{T}} = K$. Thus, $\Phi_d(\mathbf{X})$ and $\tilde{U}_K \tilde{S}_K^{1/2}$ are approximately the same, up to a unitary transformation. We need to identify different sources of errors to understand the approximation quality.

**Error source 1**. *Finite samples to learn the kernel.* We want to infer about "continuous objects" $\kappa$ and $\mathcal{D}$ (specifically the eigenfunctions of $\mathcal{K}$) but $K$ only contains the kernel values of a finite set of pairs. From standard results in Kernel PCA [52, 25], we have with probability $\geq 1 - \epsilon$,

$$\|U_K S_K^{1/2} W - \Phi_d(X)\|_F \leq 2\sqrt{2}\frac{\sqrt{\log \epsilon^{-1}}}{\lambda_d(\mathcal{K}) - \lambda_{d+1}(\mathcal{K})} = 2\sqrt{2}\frac{\sqrt{\log \epsilon^{-1}}}{\delta_d}.$$

**Error source 2**. *Only observe $A$.* We observe only the realized graph $A$ and not $K$, though it holds that $\mathbb{E}A = K/C(n)$. Thus, we can only use singular vectors of $C(n)A$ to approximate $\tilde{U}_K \tilde{S}_K^{1/2}$. We have: $\left\|\sqrt{C(n)}U_A S_A^{1/2}W - U_K S_K^{1/2}\right\|_F = O\left(\frac{t\sqrt{dn}}{\delta_d^2 \rho(n)}\right)$. When $A$ is dense (*i.e.,* $C(n) = O(1)$), the problem is analyzed in [25]. We generalize the results in [25] for the sparse graph case. See [19] for a complete analysis.

**Error source 3**. *Truncation error.* When $i$ is large, the noise in $\lambda_i(A)(\tilde{U}_A)_{:,i}$ "outweighs" the signal. Thus, we need to choose a $d$ such that only the first $d$ eigenvectors/eigenvalues of $A$ are used to approximate $\Phi_d$. Here, we need to address *the truncation error*: the tail $\{\sqrt{\lambda_i}\psi_i(x_j)\}_{i>d}$ is thrown away.

Next we analyze the magitude of the tail. We abuse notation so that $\Phi_d(x)$ refers to both a $d$-dimensional vector and a $N_{\mathcal{H}}$-dimensional vector in which all entries after the $d$-th one are 0. We have $\mathbb{E}\|\Phi(x) - \Phi_d(x)\|^2 = \sum_{i>d}\mathbb{E}[(\sqrt{\lambda_i}\psi_i(x))^2] = \sum_{i>d}\lambda_i \int |\psi_i(x)|^2 dF(x) = \sum_{i>d}\lambda_i$. (A Chernoff bound is used to obtain that $\|\Phi - \Phi_d\|_F = O(\sqrt{n}/(\sqrt{\sum_{i>d}\lambda_i}))$). Using the decay condition, we show that a $d$ can be identified so that the tail can be bounded by a polynomial in $\delta_d$. The details are technical and are provided in [19].

## 3.2 Estimating Pairwise Distances from $\widehat{\Phi}(x_i)$ through Isomap

See ISOMAP-ALGO$(\cdot)$ in Fig. 1 for the pseudocode. After we construct our estimate $\widehat{\Phi}_d$, we estimate $K$ by letting $\widehat{K} = \widehat{\Phi}_d \widehat{\Phi}_d^{\mathsf{T}}$. Recalling $K_{i,j} = c_0/(|x_i - x_j|^\Delta + c_1)$, a plausible approach is to estimate $|x_i - x_j| = (c_0/\widehat{K}_{i,j} - c_1)^{1/\Delta}$. However, $\kappa(x_i, x_j)$ is a convex function in $|x_i - x_j|$.

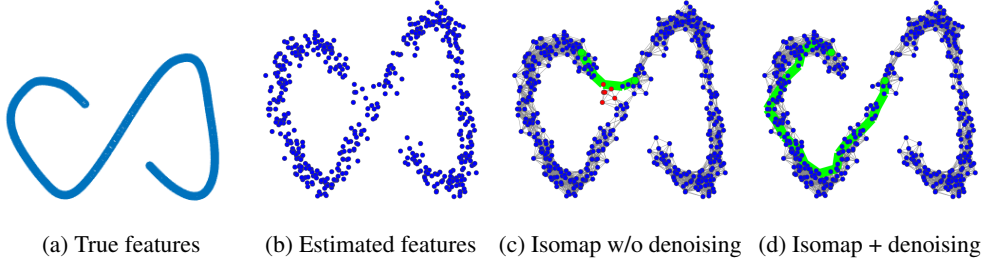

(a) True features    (b) Estimated features    (c) Isomap w/o denoising    (d) Isomap + denoising

Figure 2: Using the Isomap Algorithm to recover pairwise distances. (a) The true curve $\mathcal{C} = \{\Phi(x)\}_{x \in [0,1]}$ (b) Estimate $\widehat{\Phi}$ (c) Shows that an undesirable short-cut may exist when we run the Isomap algorithm and (d) Shows the result of running the Isomap algorithm after removal of the corrupted nodes.

Thus, when $K_{i,j}$ is small, a small estimation error here will result in an amplified estimation error in $|x_i - x_j|$ (see also Fig. 7 in App. A.3). But when $|x_i - x_j|$ is small, $K_{i,j}$ is reliable (see the "reliable" region in Fig. 7 in App. A.3).

Thus, our algorithm only uses large values of $K_{i,j}$ to construct estimates. The isomap technique introduced in topological learning [53, 54] is designed to handle this setting. Specifically, the set $\mathcal{C} = \{\Phi(x)\}_{x \in [0,1]}$ forms a curve in $\mathbb{R}^{N_{\mathcal{H}}}$ (Fig. 2(a)). Our estimate $\{\widehat{\Phi}(x_i)\}_{i \in [n]}$ will be a noisy approximation of the curve (Fig. 2(b)). Thus, we build up a graph on $\{\Phi(x_i)\}_{i \leq n}$ so that $x_i$ and $x_j$ are connected if and only if $\widehat{\Phi}(x_i)$ and $\widehat{\Phi}(x_j)$ are close (Fig. 2(c-d)). Then the shortest path distance on $G$ approximates the geodesic distance on $\mathcal{C}$. By using the fact that $\kappa$ is a radial basis kernel, the geodesic distance will also be proportional to the latent distance.

**Corrupted nodes.** Excessively corrupted nodes may help build up "undesirable bridges" and interfere with the shortest-path based estimation (cf.Fig. 2(c)). Here, the shortest path between two green nodes "jumps through" the excessively corrupted nodes (labeled in red) so the shortest path distance is very different from the geodesic distance.

Below, we describe a procedure to remove excessively corrupted nodes and then explain how to analyze the isomap technique's performance after their removal. Note that $d$ in this section mostly refers to the shortest path distance.

**Step 1. Eliminate corrupted nodes.** Recall that $x_1, x_2, ..., x_n$ are the latent variables. Let $z_i = \Phi(x_i)$ and $\widehat{z}_i = \widehat{\Phi}(x_i)$. For any $z \in \mathbb{R}^{N_{\mathcal{H}}}$ and $r > 0$, we let $\mathbf{Ball}(z, r) = \{z' : \|z' - z\| \leq r\}$. Define projection $\mathrm{Proj}(z) = \arg\min_{z' \in \mathcal{C}} \|z' - z\|$, where $\mathcal{C}$ is the curve formed by $\{\phi(x)\}_{x \in [0,1]}$. Finally, for any point $z \in \mathcal{C}$, define $\Phi^{-1}(z)$ such that $\Phi(\Phi^{-1}(z)) = z$ (*i.e.,* $z$'s original latent position). For the points that fall outside of $\mathcal{C}$, define $\Phi^{-1}(z) = \Phi^{-1}(\mathrm{Proj}(z))$. Let us re-parametrize the error term in Proposition 3.1. Let $f(n)$ be such that $\|\widehat{\Phi} - \Phi\|_F \leq \sqrt{n}/f(n)$, where $f(n) = \rho^{2/87}(n) = \Omega(\log^2 n)$ for sufficiently large $\rho(n)$. By Markov's inequality, we have $\Pr_i[\|\widehat{\Phi}(x_i) - \Phi(x_i)\|^2 \geq 1/\sqrt{f(n)}] \leq 1/f(n)$. Intuitively, when $\|\widehat{\Phi}(x_i) - \Phi(x_i)\|^2 \geq 1/\sqrt{f(n)}$, $i$ becomes a candidate that can serve to build up undesirable shortcuts. Thus, we want to eliminate these nodes.

Looking at a ball of radius $O(1/\sqrt{f(n)})$ centered at a point $\widehat{z}_i$, consider two cases.
*Case 1.* If $\widehat{z}_i$ is close to $\mathrm{Proj}(\widehat{z}_i)$, *i.e.,* corresponding to the blue nodes in Figure 2(c). For the purpose of exposition, let us assume $\widehat{z}_i = z_i$. Now for any point $z_j$, if $|x_i - x_j| = O(f^{-1/\Delta}(n))$, then we have $\|\widehat{z}_i - \widehat{z}_j\| = O(1/\sqrt{f(n)})$, which means $z_j$ is in $\mathbf{Ball}(z_i, O(1/\sqrt{f(n)}))$. The total number of such nodes will be in the order of $\Theta(n/f^{1/\Delta}(n))$, by using the near-uniform density assumption.
*Case 2.* If $\widehat{z}_i$ is far away from any point in $\mathcal{C}$, *i.e.,* corresponding to the red ball in Figure 2(c), any points in $\mathbf{Ball}(\widehat{z}_i, O(1/\sqrt{f(n)}))$ will also be far from $\mathcal{C}$. Then the total number of such nodes will be $O(n/f(n))$.

As $n/f^{1/\Delta}(n) = \omega(n/f(n))$ for $\Delta > 1$, there is a phase-transition phenomenon: When $\widehat{z}_i$ is far from $\mathcal{C}$, then a neighborhood of $\widehat{z}_i$ contains $O(n/f(n))$ nodes. When $\widehat{z}_i$ is close to $\mathcal{C}$, then a neighborhood of $\widehat{z}_i$ contains $\omega(n/f(n))$ nodes. We can leverage this intuition to design a counting-based algorithm to eliminate nodes that are far from $\mathcal{C}$:

$$\textsc{Denoise}(\widehat{z}_i) : \text{If } |\mathbf{Ball}(\widehat{z}_i, 3/\sqrt{f(n)})| < n/f(n), \text{ remove } \widehat{z}_i. \tag{3}$$

| Algo. | $\rho$ | Slope of $\beta$ | S.E. | p-value |
|---|---|---|---|---|
| Ours | **0.53** | 9.54 | 0.28 | $< 0.001$ |
| Mod. [55] | 0.16 | 1.14 | 0.02 | $< 0.001$ |
| CA [18] | 0.20 | 0.11 | 7e-4 | $< 0.001$ |
| Maj [56] | 0.13 | 0.09 | 0.02 | $< 0.001$ |
| RW [54] | 0.01 | 1.92 | 0.65 | $< 0.001$ |
| MDS [49] | 0.05 | 30.91 | 120.9 | 0.09 |

Figure 3: Latent Estimates vs. Ground-truth.

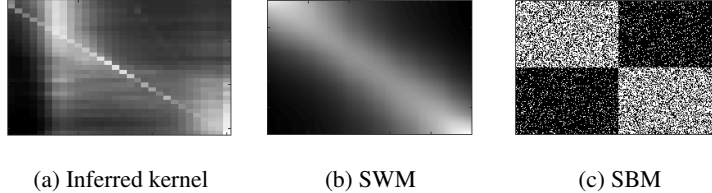

     (a) Inferred kernel        (b) SWM        (c) SBM

Figure 4: Visualization of real and synthetic networks. (a) Our inferred kernel matrix, which is "in-between" (b) the small-world model and (c) the stochastic blockmodel.

*Theoretical result.* We classify a point $i$ into three groups:

1. **Good**: Satisfying $\|\widehat{z}_i - \mathrm{Proj}(\widehat{z}_i)\| \leq 1/\sqrt{f(n)}$. We further partition the set of good points into two parts. Good-I are points such that $\|\widehat{z}_i - z_i\| \leq 1/\sqrt{f(n)}$, while Good-II are points that are good but not in Good-I.
2. **Bad**: when $\|z_i - \mathrm{Proj}(z_i)\| > 4/\sqrt{f(n)}$.
3. **Unclear**: otherwise.

**Lemma 3.2.** *(cf. [19] ) After running* DENOISE *that uses the counting-based decision rule, all good points are kept, all bad points are eliminated, and all unclear points have no performance guarantee. The total number of eliminated nodes is $\leq n/f(n)$.*

**Step 2. An isomap-based algorithm**. Wlog assume there is only one closed interval for $\mathrm{support}(F)$. We build a graph $G$ on $[n]$ so that two nodes $\widehat{z}_i$ and $\widehat{z}_j$ are connected if and only if $\|\widehat{z}_i - \widehat{z}_j\| \leq \ell/\sqrt{f(n)}$, where $\ell$ is a sufficiently large constant (say 10). Consider the shortest path distance between arbitrary pairs of nodes $i$ and $j$ (that are not eliminated.) Because the corrupted nodes are removed, the whole path is around $\mathcal{C}$. Also, by the uniform density assumption, walking on the shortest path in $G$ is equivalent to walking on $\mathcal{C}$ with "uniform speed", *i.e.,* each edge on the path will map to an approximately fixed distance on $\mathcal{C}$. Thus, the shortest path distance scales with the latent distance, *i.e.,* $(d-1)\left(\frac{c}{2}\right)^{1/\Delta}\left(\frac{\ell-3}{\sqrt{f(n)}}\right)^{2/\Delta} \leq |x_i - x_j| \leq d\left(\frac{c}{2}\right)^{1/\Delta}\left(\frac{\ell+8}{\sqrt{f(n)}}\right)^{2/\Delta}$, which implies Theorem 2.2 (cf. [19] for details).

**Discussion: "Gluing together" two algorithms?** The unified model is much more flexible than SBM and SWM. We were intrigued that the generalized algorithm needs only to "glue together" important techniques used in both models: Step 1 uses the spectral technique inspired by SBM inference methods, while Step 2 resembles techniques used in SWM: the isomap $G$ only connects between two nodes that are close, which is akin to throwing away the long-range edges.

## 4 Experiments

We apply our algorithm to a social interaction graph from Twitter to construct users' ideology scores. We assembled a dataset by tracking keywords related to the 2016 US presidential election for 10 million users. First, we note that as of 2016 the Twitter interaction graph behaves "in-between" the small-world and stochastic blockmodels (see Figure 4), *i.e.,* the latent distributions are bi-modal but not as extreme as the SBM.

*Ground-truth data.* Ideology scores of the US Congress (estimated by third parties [57]) are usually considered as a "ground-truth" dataset, *e.g.,* [18]. We apply our algorithm and other baselines on Twitter data to estimate the ideology score of politicians (members of the 114th Congress), and

observe that our algorithm has the highest correlation with ground-truth. See Fig. 3. Beyond correlation, we also need to estimate the statistical significance of our estimates. We set up a linear model $y \sim \beta_1 \widehat{x} + \beta_0$, in which $\widehat{x}$'s are our estimates and $y$'s are ground-truth. We use bootstrapping to compute the standard error of our estimator, and use the standard error to estimate the p-value of our estimator. The details of this experiment and additional empirical evaluation are available in [19].

**Acknowlegments**

The authors thank Amazon for partly providing AWS Cloud Credits for this research.

## Footnotes

*Currently at Google.

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
