[Supplementary Material · supp.pdf]

# A  Additional Illustrations

This section provides additional illustrations related to our work.

## A.1  Model Selection Probem (presented in Section 1)

Without observing the latent positions or knowing which model generated the underlying graph, the adjacency matrix of a social graph typically looks like the one shown in Fig. 5(a). However, if the model generating the graph is known, it is then possible to run a suitable "clustering algorithm" [14, 16] that reveals the hidden structure. When the vertices are ordered suitably, the SBM's adjacency matrix looks like the one shown in Fig. 5(b) and that of the SWM looks like the one shown in Fig. 5(c).

**(a) Input**     **(b) Stochastic Block Model**     **(c) Small World Model**

Figure 5: Inference problem in social graphs: Given an input graph (a), are we able to shuffle the nodes so that statistical patterns are revealed? A major problem in network inference is the model selection problem. The input here can come from stochastic block model (b) or small-world model (c).

## A.2  Algorithm for the Small-world Model (presented in Section 2)

The inference algorithm for small-world networks uses different ideas. Each edge in the graph can be thought of as a "short-range" or "long-range" one. Short-range edges are those between nodes that are nearby in latent space, while long-range ones have end-points that are far away in latent space. After the removal of all the long-range edges, the shortest path distance between two nodes scales proportionally to the corresponding latent space distance (see Fig. 6). Once estimates for pairwise distances are obtained, standard buidling blocks may be used to find the latent positions $x_i$ [40].

Figure 6: In the small-world model, after removal of long range edges (red thick), the shortest-path distance between two nodes approximates latent space distance

## A.3  Sensitivity of the Gram matrix $K$ (presented in Section 3.2)

After we construct our estimate $\widehat{\Phi}_d$, we may estimate $K$ by letting $\widehat{K} = \widehat{\Phi}_d \widehat{\Phi}_d^{\mathsf{T}}$. Recalling $K_{i,j} = c_0/(|x_i - x_j|^\Delta + c_1)$, one plausible approach would be estimating $|x_i - x_j| = (c_0/\widehat{K}_{i,j} - c_1)^{1/\Delta}$. A main issue with this approach is that $\kappa(x_i, x_j)$ is a convex function in $|x_i - x_j|$. Thus, when $K_{i,j}$ is small, a small estimation error here will result in an amplified estimation error in $|x_i - x_j|$ (cf. Fig. 7). But when $|x_i - x_j|$ is small, $K_{i,j}$ is reliable (see the "reliable" region in Fig. 7).

Figure 7: The behavior of $\widehat{K}_{i,j}$.

## B  Summary of notations

- $A \in R^{n \times n}$: the adjacent matrix of an undirected graph. In the simplified graph model, $A$ is the input graph. In the bipartite graph model, $A$ is the regularized matrix over $B^{\mathsf{T}}B$.
- $B \in R^{m \times n}$: the bipartite graph matrix, *i.e.*, $B_{i,j} = 1$ if and only if follower $i$ is connected to influencer $j$.
- **Ball**$(z, r) = \{z' : \|z' - z\| \leq r\}$
- $C(n)$: the normalization constant in the undirected graph model, *i.e.*, $\Pr[\{x_i, x_j\} \in E] = \kappa(x_i, x_j)/C(n)$.
- $\mathcal{C}$: a curve in $R^{N_{\mathcal{H}}}$, defined as $\mathcal{C} = \{\Phi(x)\}_{x \in [0,1]}$
- $\mathcal{D}$: the distribution in which $x_i$ and $y_I$ come from.
- $D$: the distance estimate, *e.g.*, $D_{i,j}$ is the estimate of distance between $x_i$ and $x_j$ in our algorithm.
- $d$: the number of eigenvalues to keep; in the section for isomap technique, it sometimes is used to refer to the length of the shortest path.
- $\mathcal{H}$: the reproducing kernel Hilbert space of $\kappa$.
- $K \in R^{n \times n}$: the kernel/Gram matrix associated with $\kappa$, *i.e.*, $K_{i,j} = \kappa(x_i, x_j)$.
- $\mathcal{K}$: an integral operator defined as $\mathcal{K}f(x) = \int \kappa(x, x')f(x')dF(x')$.
- $F(n)$: the cdf of $\mathcal{D}$.
- $\mathcal{M}$: an integral operator defined as $\mathcal{M}f(x) = \int \mu(x, y)f(y)dF(y)$.
- $M \in R^{n \times n}$: the kernel/Gram matrix associated with $\mu$, *i.e.*, $M_{i,j} = \mu(x_i, x_j)$.
- $m$: the number of followers in the bipartite graph model.
- $N_{\mathcal{H}}$: the number of eigenvalues in $\mathcal{K}$, which could be countably infinite.
- $n$: the number of nodes in the simplified model and the number of influencers in the bipartite grpah model.
- $\mathcal{P}$: projection operators.
- $[\tilde{U}_X, \tilde{S}_X, \tilde{V}_S]$ where $X \in \{A, K, M\}$: the SVD of $X$.
- $[U_X, S_X, V_S]$, where $X \in \{A, K, M\}$: the first $d$ singular vectors/values of $X$. Note here they implicitly depend on $d$.
- $\mathbf{X} = \{x_1, ..., x_n\}$: the set of nodes in the simplified model and the set of influencers in the bipartite graph model.
- $\widehat{x}_i$: our algorithm's estimate of $x_i$.
- $\mathbf{Y} = \{y_1, ..., y_n\}$: the set of followers in the bipartite graph model.
- $\widehat{y}_i$: our algorithm's estimate of $y_i$.
- $z_i$: the feature of $x_i$, *i.e.*, $\Phi(z_i)$.

- $\widehat{z}_i$: our algorithm's estimate of $z_i$.
- $\delta_i$: the eigengap, defined as $\delta_i = \lambda_i - \lambda_{i+1}$.
- $\lambda_i$: the eigenvalues of $\mathcal{K}$ unless otherwise specified.
- $\rho(n)$: we also re-parametrize $C(n) = n/\rho(n)$, *i.e.,* $\rho(n) = n/C(n)$.
- $\kappa : [0,1] \times [0,1] \to (0,1]$: the kernel function.
- $\Delta$: the exponent in the small-world kernel, *i.e.,* $\kappa(x_i, x_j) = c_0/(|x_i - x_j|^\Delta + c_0$.
- $\Phi : [0,1] \to R^{N_{\mathcal{H}}}$ the feature map associated with $\mathcal{K}$.
- $\widehat{\Phi}$: our algorithm's estimate of $\Phi$.
- $\Phi^{\mathcal{M}}$: the feature map associated with $\mathcal{M}$.
- $\Phi_d$: the first $d$ coordinates of the feature map. It is also overloaded to be in $R^{N_{\mathcal{H}}}$ by padding 0's after the $d$-th coordinate.
- $\widehat{\Phi}_d$: our algorithm's estimate of $\Phi_d$.
- $\Phi_d^{\mathcal{M}}$: the first $d$ coordinates of $\Phi^{\mathcal{M}}$.
- $\psi_i$: the $i$-th eigenfunction of $\mathcal{K}$.