[Reviews · NeurIPS 2017]

Reviewer 1



(I have been asked by the area chairs to provide an extra set of eyes for this paper. Because this review is entering the discussion after the rebuttal phase, discussants were asked to consider anything raised here in light of this lack of an opportunity to give a rebuttal, as it is possible that I have misunderstood things that a rebuttal would have addressed.) The work presents itself as contributing a unification of two widely studied random graph models, the stochastic block model and small world model, and developing an efficient algorithm for recovering latent representations for this unified model. A problem with this claim is that, as is acknowledged in the work, this “new” unified approach is really just a (sampled) graphon model, as the authors make clear on Lines 113-114. It’s plain to any reader familiar with graphons that adaptations of both the SBM and small world model fit within the graphon framework. The SBM-graphon connection is obvious and well-studied; the SW-graphon connection is obvious in-so-much-as the small world model is really just a latent space model, though most attention for latent space graphons has been on e.g. logistic latent space models. Denoting that model the "small world” model invokes expectations of navigability results, so in general the results would be better explained in terms of a latent space model with an inverse polynomial distance kernel. All that said, the work is considered on it's merits as a contribution to the graphon literature. Detailed comments: 1) The way the paper presents it’s results does a disservice to their strengths. The paper contributes a careful analysis of a complex algorithm that appears to bring earlier results for sampled dense graphons over to sparse graphons (Line 255: "We generalize the results in [24] for the sparse graph case.”) That said, while the results in [24] are explicitly given for dense graphs, they are sketched there for the sparse case as well (in the conclusions, with the comment "A precise statement and formulation of the results in Section 4 for the sparse setting might require some care, but should be for the most part straightforward.”). It is possible that the details of the sketch in [24] are not nearly as “straight-forward” as the authors there claim. But engaging in a clear dialogue with the literature is a reasonable requirements for acceptance, in the eyes of this reviewer. If the paper had a (1) title and abstract and (2) related work section that made clear the position of the paper in the sampled graphon literature, specifically related to [24], that would _greatly_ improve the ability of a reader to appreciate the work. As is, the results pertaining to recovering both SBM and SWM networks claimed in the abstract and introduction are essentially results achieved by [24], except for (a) the sparse graph case (the polynomial small world kernel is a universal kernel in the language of the results of that work) and (b) the pairwise distance recovery. I will now focus on the pairwise distance recovery. Basically, [24] is quite general, as is the core of this work. And the focus of the title and introduction on mis-specification between the SBM and SWM feels like the wrong problem to focus on when putting forward this work. 2) Connecting the extended analysis of [24] -- roughly, to get the kernel PCA to work in the sparse regime -- is a nice if modest contribution. More significant is connecting the output of this machinery to MDS machinery to obtain the pairwise distances. This appears to be a novel point, and it seems quite useful. The analysis here does rely on some specific details of the small-world kernel, but it would be really helpful if the authors clarified what the general conditions are on the latent space kernel for some version of pairwise distance recovery to go through. It seems that the long-range links allowed by the heavy polynomial tails may be critical; is that so? Do any version of the results work for kernels with exponential tails? 3) In line with the odd focus of the introduction, the empirical comparisons made in the work are a bit off, but I'm willing to let this slip as the work isn't really an empirical paper, and should get positive "credit" for even trying to do empirical work here. The empirical work focuses on comparisons with algorithms that aren’t quite trying to do what the present algorithm is trying to do, while not benchmarking itself against methods that are designed for some of the same tasks. First, the SBA algorithm of Airoldi-Costa-Chan and SAS algorithm of Chen-Airoldi, and related network histogram methods are specifically designed for the estimation problem in Figure 4. But not compared. Second, the methods used in comparison are not really reasonable. Majority vote label propagation from [56] is an algorithm for categorical labels, not continuous embeddings. It’s not clear why [54], a paper about conformal Isomap, is called a random walk label propagation method, so that might be a crossed-up citation. It is likely that the modularity method [55] is finding a core-periphery cut with it’s main eigenvector; for this reason it would be interesting to see how a semi-supervised method does (e.g. ZGL, given the labels for the presidential candidates discussed as being given to other methods). As an overall comment, the empirical question is a terrific one, with high quality data, but the evaluation is not up to the standards of a thorough empirical investigation. But again, I realize that the main contribution of this work is theoretical, but I still want to register what I see as issues. I am most concerned about the lack of comparison to the other methods in the estimation in Figure 4 and Figure S9. 3) As a small comment about Figure (S)9: for the methods under consideration it would be more honest to show the performance of the Abraham et al. algorithm in Fig 9d and the Newman spectral algorithm in Fig 9b, as opposed to just showing the cross-objectives (where the two methods fail). 4) One point that confused this reader for some time was the fact that the paper studies sparse graphons without any discussion of the tricky issues for such models. It should be made very clear that the model under consideration is the Bollobas-Janson-Riordan (2007) model (also studied by Bickel-Chen-Levina (2011), which is a sampled dense graphon. Some pointers to the emerging literature on sparse graphons defined through edge-exchangeable ideas (Crane-Dempsey 2016 building on Caron-Fox 2014, Veitch-Roy 2015, Borgs et al. 2016) would be useful to provide. 5) The related work section overlooks that [22], Airoldi et al., makes computational improvements (in terms of complexity) on the USVT method (but still for dense graphs). 6) Ref [19], the latent space model of Hoff et al., is pegged as a “Graphon-based technique” where it should be raised as a broad citation for latent space models such as the small world model discussed herein. 7) The discussion of results for both the unipartite and bipartite graph model is nice. The bipartite model discussed here is sometimes called an affiliation network, see "Affiliation networks” by Lattanzi and Sivakumar, perhaps worth connecting to. I think there are a lot of interesting connections made in the work. The management of errors in the isomap algorithm for recovering latent space graph models is a nice contribution.

Reviewer 2



Summary of paper: The paper first proposes a unified model for graphs that generalises both stochastic block model (SBM) and small-world model (SWM). Two versions of the unified model are presented, and algorithm and analysis are given for estimating the models. -> Model 1 (Simplified model): A minor modification of the sparse graphon as in [R1, see below]. Here, one would sample n points x_1, … , x_n from [0,1] according to some distribution D (that may not be uniform), and then probability is given by C(n)*k(x_i,x_j), where C controls sparsity and k is the kernel function that is assumed have a specific form. -> Model 2 (Bipartite model): This is essentially a generalisation of sparse graphon to bipartite graphs. Here, one samples x_1, … , x_n ~ D on [0,1] and also y_1, … , y_m ~ D on [0,1], and in the graph, edges only occur between x-nodes and y-nodes with probability C(n)*k(x_i,y_j). This model is presented in main text, but estimation is given in appendix. Algorithms for estimating the models are given and analysed. Main idea is to first get a spectral embedding from adjacency matrix, then apply Isomap to get distance matrix from embedding. Finally, apply line embedding algorithm to get x_i’s from distances. The first two steps are analysed and the last is assumed to work well based on existing results. I have not checked all the proofs thoroughly since most intermediate claims seem intuitively correct to me. Pros: 1. The algorithm is somewhat novel though its individual components are well known. While most works end at graphon kernel estimation, this paper estimate distances under assumed structure of kernel. Some parts of proof also seem novel, but I am not very familiar with the literature on Isomap. 2. There is limited work on estimating sparse bipartite graphon. Cons: 1. The simplified model can be derived from sparse graphon [R1] and is same as [29]. Also density regime is not very sparse. So spectral embedding analysis has limited novelty. 1. 2. The bipartite model is interesting, but not entirely new. Bipartite graphs have been studied in the context of partitioning [R2,R3], and dense bi-partite graphons are also known [R5]. Some techniques for bipartite model is somewhat similar to [R2]. 3. The main paper is written in a very confusing way (particularly Section 2), and many interesting parts and potential contributions can be found only in Appendix. Detailed concerns: Line 81: Recovery is possible when p,q = Omega(log(n)/n), not polylog as claimed in the paper. Polylog(n)/n is an easier regime where trace method type bounds are applicable (see [R4]). [5,9] study log(n)/n case. This claim repeats throughout the paper, and the authors actually use much simpler techniques for spectral embedding. For sparse graphs, one needs to consider regularisation of the adjacency matrix [R4]. Line 103-126: It took me long time to realise that the paper considers two very different estimation problems. A reader normally would imagine everything goes into making of a big graph with both influencers and followers. The paper has no clearly stated problem definition. Line 118-126: This model follows from sparse graphon [R1]. Let F be the cumulative distribution of D. Then x_i = F^{-1}(z_i) for z_i ~ Unif[0,1]. So we can define the kernel in terms of z_i which is sparse graphon model. Similar argument can also be used for bipartite case. Line 140-141: For bipartite graph, [R2] also studies decomposition of B’B. They suggest a regularisation by deleting the diagonal, which is quite similar to the diagonal diminishing approach used here. Though [R2] only studied partitioning, it allowed more disparity between m and n, and more sparse regime. Line 199-203: Though line embedding is used as a blackbox, its effect on the overall guarantee must be accounted for. Also, do results of [48,49] hold when distances are known upto some error? Figure 1: Do we assume \rho(n) is given? Shouldn’t this be estimated as well? If \rho(n) is assumed known, then results of [R1] are applicable. Line 348-366: This section, and the main paper in general fails to motivate why is it more important to estimate latent position instead of graphon kernel. For instance, Figure 4 still shows only the kernel matrix -- then why shouldn’t we use the more general theory of graphon estimation? The only motivation can be found at the end in Appendix G. Line 324-332: No guarantee is known for unclear points. Why doesn’t this cause any problem in this part? Maybe, I just missed this point in the proof. So a pointer would suffice. Line 64: USVT works in the polylog(n)/n regime. See proof in [28]. So in the assumed density regime, one could also use USVT instead of estimating d via DecideThreshold. Line 36-39,99: It is quite inconvenient if a paper refers to figures in appendix for introductory discussion. [R1] Klopp, Tsybakov, Verzelen. Oracle inequalities for network models and sparse graphon estimation. [R2] Florescu, Perkins. Spectral Thresholds in the Bipartite Stochastic Block Model. [R3] Feldman, Perkins, Vempala. Subsampled Power Iteration: a Unified Algorithm for Block Models and Planted CSP’s. [R4] Le, Vershynin. Concentration and regularization of random graphs. [R5] Choi. Co-clustering of nonsmooth graphons.

Reviewer 3



This paper studies the latent position model for networks. The authors assume a small world kernel in the latent space. By combining techniques like spectral analysis and isomap, they are able to provide a provable procedure to recover the latent structure of the nodes in the graph. They illustrate their method on a twitter dataset for political ideology estimation. Overall I think the paper is well-written, the theory looks correct and proof ideas are provided which makes it very easy to follow. Below are some detailed comments and questions. 1. The algorithm and analysis can be broken down into two parts, first is recovering the low rank feature map, and the second is to control the error in isomap. The analysis for the first part is standard, and similar to those in [24]. Also this part could be very general, it should hold for many other latent position models such as random dot product graph etc. The second part of the analysis is tailored to the small world kernel and seems novel. It would be nice to see some discussions about how this technique generalizes to other type of kernels. 2. I have some confusion making sense of the bipartite model assumption. I understand that one wants to divide the nodes into high-degree and low-degree parts, but why is it necessary to assume that there is no connection within high degree nodes? In most real world networks the celebrity nodes also have edges within themselves. In fact, it seems to me having more inter-cluster edges would only help the inference. Is this an assumption that mainly for theoretical convenience or can the author provide more insights in the rebuttal? 3. The authors criticize the degree-corrected block model on the fact that, "the subgraph induced by the high degree nodes is dense, which is inconsistent with real-world networks" (line 146-147). However if I understand correctly, Proposition 3.1 also only holds for dense graphs (\rho(n)=\omega(\log n)), which means for the simplified model, a graph consists of all celebrities is dense. (just to make sure we are on the same page, I'm referring to sparse when the average degree is O(1)). How is this different from DC-SBM? I failed to see the sparsity claim the authors made in the abstract (line 9) and line 255. Please clarify. 4. The authors claim that they can make a distinction between SBM and SWM, but the main theorem only talks about the recovery of the pairwise distances. I was wondering if someone would like to do a model selection, for instance, to figure out whether a SBM algorithm applies, how the authors would suggest him to do based on the inferred latent positions. In other words, is there any procedure that tells me whether the latent distribution is discrete or continuous, up to a certain accuracy? 5. The experiments provides very interesting insight on this dataset. It is very impressive that the authors manually labeled the data for classification task. For G 2.1 it is seems that majority with modularity initialization does a fairly good job, but as said in the paper, the proposed approach delivers more information on the finer granularity than a binary labeling, which is nice. 6. It is important to make the paper self-contained. For instance, line 36 talks about figures in the supplementary, and figure 8 in theorem 2.2 is actually not in the main paper. Also as I mentioned, I like the fact that the authors put a lot of intuition for the proof in the main paper, but the current layout is a bit too crowded, there's barely any space under caption of figure 2 to distinguish it from the text. I would suggest moving more things to supplementary to improve readability. 7. Line 849 requires some elaboration. Other minor issues: typo in line 159 "involved" in line 250-251, the W is undefined, might be some unitary matrix I assume? line 292, the curve is defined by \Phi capitalized?